# Identification and detection of genetic markers associated with antimicrobial susceptibility and evaluation of efflux pump mechanisms in *Mycoplasma iowae*

Dominika Buni[1], Áron Botond Kovács[1,2], Enikő Wehmann [1], Dénes Grózner [1,3,4], Krisztián Bányai[1,5], Eszter Zsófia Nagy[1,3], Janet Bradbury[6], Marco Bottinelli[7], Elisabetta Stefani[7], Salvatore Catania[7], Inna Lysnyansky[8], László Kovács[5,9], Miklós Gyuranecz[1,3,4,5,10], Zsuzsa Kreizinger [1,3,4,10]*

1 HUN-REN Veterinary Medical Research Institute, Budapest, Hungary, 2 Institute of Metagenomics, University of Debrecen, Debrecen, Hungary, 3 National Laboratory of Infectious Animal Diseases, Antimicrobial Resistance, Veterinary Public Health and Food Chain Safety, Budapest, Hungary, 4 National Laboratory of Health Safety, Budapest, Hungary, 5 University of Veterinary Medicine Budapest, Budapest, Hungary, 6 Institute of Infection, Veterinary and Ecological Sciences, University of Liverpool, Neston, Wirral, United Kingdom, 7 Istituto Zooprofilattico Sperimentale delle Venezie, Buttapietra, Verona, Italy, 8 Department of Avian Diseases, Kimron Veterinary Institute, Beit Dagan, Israel, 9 Poultry-Care Kft., Újszász, Hungary, 10 MolliScience Kft., Biatorbágy, Hungary

* kreizinger.zsuzsa@vmri.hun-ren.hu

## Abstract

*Mycoplasma iowae* is an economically significant pathogen that causes reduced hatchability, late embryo mortality and leg deformities, chondrodystrophy and skeletal lesions in poults. While prevention is essential in the control of infection, the appropriate administration of antibiotics may reduce economic losses during outbreaks. As a first step in the exploration of antimicrobial resistance mechanisms in *M. iowae*, target modification and efflux pump activity were examined in the present study. Point mutations were analyzed in previously described antibiotic binding sites in the whole genome sequences of 99 *M. iowae* strains. Mismatch amplification mutation assays (MAMAs) were designed and validated for the differentiation of mutations corresponding to elevated minimum inhibitory concentration (MIC) values for fluoroquinolones. Broth microdilution assays were performed to evaluate the effect of efflux pump inhibitors. In the presence of orthovanadate (OV), MIC values were significantly lower than in the absence of OV for spiramycin, tilmicosin, tylosin and oxytetracycline, which may indicate the presence of an active efflux system in *M. iowae*. Putative promoter regions of efflux-related genes were predicted and characterized. Genetic mutations, previously described in other bacteria, were described to be associated with elevated fluoroquinolone, macrolide and lincomycin MICs in *M. iowae*, although certain resistant phenotypes remained unexplained, promoting future examinations for deeper insights. The developed MAMAs may support rapid identification of *M. iowae* strains with elevated MIC values for fluoroquinolones. The better

**Data availability statement:** All whole-genome sequence data are available from GenBank (BioProject ID: PRJNA975348, GenBank accession numbers: LR215023 and CP033512.2.1). All other relevant data are within the manuscript and its Supporting Information files.

**Funding:** This work was supported by the FK21 (137809) grant of ZK from the National Research, Development and Innovation Office, Hungary; the SA-27/2021 grant of MG from the Eötvös Loránd Research Network; the Projects no. RRF-2.3.1-21-2022-00001 and RRF-2.3.1-21-2022-00006 of MG which have been implemented with the support provided by the Recovery and Resilience Facility (RRF), financed under the National Recovery Fund budget estimate,RRF-2.3.1-21 funding scheme; and the support of MG provided by the Ministry of Innovation and Technology of Hungary (legal successor: Ministry of Culture and Innovation of Hungary) from the National Research, Development and Innovation Fund, financed under the TKP2021-EGA-01 funding scheme of the National Research, Development and Innovation Office.

**Competing interests:** The authors have declared that no competing interests exist.

understanding of the efflux pump mechanisms enables the development of alternative methods for the support of therapy against this pathogen.

## Introduction

*Mycoplasma iowae* is an economically significant pathogen that causes reduced hatchability, late embryonic mortality and leg deformities, chondrodysplasia and skeletal lesions. It mainly affects turkeys but it has been described in chickens, geese, game and exotic birds [1–5]. The absence of a vaccine against this pathogen and the fact that *M. iowae* is generally more resistant to antimicrobials than other avian pathogenic mycoplasmas [6,7] make the control of the disease complicated. The determination of the antibiotic susceptibility of mycoplasmas is a time-consuming process. Furthermore, there is a lack of standard quality control strains and defined breakpoints for the interpretation of antibiotic susceptibility in avian pathogen mycoplasmas [8,9]. Single nucleotide polymorphisms (SNPs) associated with high minimum inhibitory concentration (MIC) values represent appropriate genetic markers for the development of new molecular techniques capable of the rapid detection of antibiotic susceptibility, hence supporting targeted antimicrobial therapy. Previous studies have described SNPs affecting macrolide and lincosamide resistance in the 23S rRNA and L22 protein genes, while for fluoroquinolones resistance markers have been found in the *gyrA*, *gyrB*, *parC* and *parE* genes in *Mycoplasma agalactiae*, *M. bovis*, *M. gallisepticum*, *M. hyopneumoniae* and *M. synoviae* [10,11]. Numerous mismatch amplification mutation assays (MAMAs) targeting these SNPs have been developed in other mycoplasmas, including *M. anserisalpingitidis*, *M. bovis*, *M. gallisepticum*, *M. hyopneumoniae*, *M. hyorhinis* and *M. synoviae* [12–16].

Efflux pumps have a key role in multidrug resistance, facilitating the elimination of a diverse range of antibiotics and chemicals from the cell. The efflux pump activity is also considered to be important in the beginning of resistance development, reducing the antibiotic concentrations in the bacterial cells until genetic mutations occur [17]. The antibiotic efflux mechanism was first identified in 1980. Since then, it has been demonstrated that the vast majority of antibiotics are prone to efflux-mediated resistance and the antibiotic efflux can be facilitated by multiple pumps within a single organism [18]. Efflux pumps are proteins located on the bacterial cell membrane. They facilitate the transport of harmful substances from within the bacterial cell to the surrounding environment. Efflux pumps have been identified in almost all bacterial species, and the genes encoding them can either be found within the bacterial genome or on plasmids [19]. The bacterial multidrug efflux transporters can be classified in two major families according to bioenergetic and structural features: primary and secondary transporters. ATP binding cassette (ABC) is a primary transporter, hydrolyzing ATP as a source of energy. Secondary transporters use proton or sodium gradient as energy source. Small multidrug resistance (SMR), major facilitator superfamily (MFS), resistance nodulation cell division (RND) and multidrug and toxic compound extrusion (MATE) pumps belong to the latter class [20,21]. The activity of efflux pumps can be inhibited at various levels, including disruption of their assembly

and expression [22]. Efflux pump inhibitors (EPIs) are molecules that inhibit efflux pumps, resulting in the inactivation of drug transport. In previous studies, three EPIs have been shown to be effective in mycoplasmas: the proton motive force (PMF) disrupter carbonyl cyanide m-chlorophenylhydrazine (CCCP); an inhibitor of MFS, RND and ATP-dependent efflux pumps, reserpine (RSP); and the ATP-dependent efflux pump inhibitor sodium orthovanadate (OV) [23–25]. As a proton ionophore, CCCP can disable energy-dependent efflux systems through alteration of the proton concentration gradient, which restores drug susceptibility by promoting drug accumulation in the cell [26–28]. RSP is a plant alkaloid that reduces the excretion of drugs in bacteria and reestablishes drug sensitivity by decreasing the hydrolysis-driven energy supply [26]. OV is a phosphate analogue, inhibiting ATPase activity [29,30]. Given the importance of efflux in the development of antibiotic resistance, EPIs can be used as adjuncts to antibiotics to enhance their activity against bacteria expressing efflux pumps, making them possible therapeutic agents [26]. The role of mutations in the promoters and upstream regions of several efflux pump genes has been described in relation to antibiotic resistance [31,32] and may contain different types of genetic events (e.g., insertions, deletions, SNPs), which can affect gene expression [32].

Previously, a total of 99 *M. iowae* strains were submitted to MIC testing with ten different antibiotics [33]. The present study aimed to identify genetic markers associated with strains exhibiting elevated MICs and to develop rapid, cost-effective molecular assays for their differentiation. In addition, the role of efflux pumps in the adaptive response to antibiotic pressure and their genetic background in *M. iowae* strains with elevated MICs was investigated. The reported findings and new molecular assays advance veterinary practice and our understanding of resistance mechanisms in this species, including the described and hypothesized characteristics.

## Materials and methods

### Samples

A total of 99 *M. iowae* clinical isolates and strains were selected for the analyses, including the type strain (NCTC 10185, strain Iowa 695) and reference strains of serotypes I (strain Iowa 695), R (D2497), N (PHN-D13), J (693), Q (L3-10) and K (1805). The NCTC 10185 type strain (Iowa 695) and the serotype I strain (Iowa 695) were obtained from different laboratories. The origin and collection date of the isolates were described previously [33] and are presented in S1 Table. The culturing of the isolates and the MIC testing have been described elsewhere [33]. One modification was applied in the current study: the isolates were cultured in MolliScience General Mycoplasma Liquid Media (MolliScience, Biatorbágy, Hungary). To examine the activity of efflux pumps, six strains each were selected which showed slightly, moderately or highly elevated MIC values previously in case of enrofloxacin, lincomycin, oxytetracycline, spiramycin, tylosin or tilmicosin (S2 Table).

### Identification of SNPs

The previously determined whole genome sequences [34] of a total of 99 *M. iowae* clinical isolates and strains (BioProject ID: PRJNA975348) were analyzed in Geneious Prime software (version 2022.2.2) [35]. Raw reads gained from Illumina NextSeq 500 platform (Illumina Inc., San Diego, CA, USA) were mapped to the *M. iowae* type strain NCTC 10185 (GenBank: LR215023.1) as reference, and examined regions were extracted from the alignments. Based on the literature, SNPs correlating with elevated MIC values were searched in the *gyrA*, *gyrB*, *parC*, *parE* genes for fluoroquinolones, and in the coding regions of the 23S rRNA, and 50S ribosomal proteins L4 and L22 for macrolides and lincosamides, [11]. The strength of association among MIC values and the identified SNPs was examined by logistic regression and Fisher's exact test with the help of online tools (Statistics Kingdom [36] and www.socscistatistics.com, respectively). For the logistic regression analyses MIC values equal or lower than the lowest examined concentrations were adjusted to be the lowest examined concentrations, while MIC values higher than the highest examined concentrations were adjusted to be two-fold higher than the highest examined concentrations. Furthermore, in order to normalize data distribution (gain linear

intervals) for the regression analysis the adjusted MIC values from the two-fold dilution series of antimicrobial concentrations were transformed to $\log_2$MIC values and were used as the predictor parameter [37,38]. The examined nucleotide positions were assigned as wild-type (0) or mutant genotypes (1) as outcome parameter (S3 Table). Contingency tables for the Fisher's exact test were assembled from the number of isolates with low or high MIC values showing the wild-type or mutant genotypes (S3 Table). MIC categories were determined as follows: "low" for strains with MIC values of ≤0.5 µg/ml for enrofloxacin, ≤ 1 µg/ml for erythromycin, ≤ 2 µg/ml for spiramycin, ≤ 8 µg/ml for tilmicosin, ≤ 1 µg/ml for tylosin, ≤ 1 µg/ml for lincomycin; "high" for strains with MIC values of >0.5 µg/ml for enrofloxacin, > 1 µg/ml for erythromycin, > 2 µg/ml for spiramycin, > 8 µg/ml for tilmicosin, > 1 µg/ml for tylosin, > 1 µg/ml for lincomycin. The positions of the resistance associated nucleotides are numbered according to the *Escherichia coli* strain K-12 substrain MG1655 (GenBank accession number: NC_000913) and *M. iowae* type strain NCTC 10185 (GenBank: LR215023.1) as recommended by the MyMIC Consortium [39].

## Development of MAMAs

MAMAs were developed to detect SNPs associated with elevated fluoroquinolone MIC values. Briefly, MAMA is a molecular tool for SNP discrimination in bacteria. The assay is based on competing allele-specific primers. These primers are SNP-specific at the 3′ end and contain a single destabilizing mismatch at the antepenultimate position of the 3′ end to increase the discriminatory power. One of the allele-specific primers is labelled with an additional 15–20 bp GC clamp. This increases the melting temperature and also the size of the amplicon. In the presence of an intercalating fluorescent DNA dye, the temperature shift can be detected on a real-time PCR platform [14,40]. All MAMA primers were designed and tested with the *M. iowae* type strain (NCTC 10185) and the 98 isolates. The melting temperature of the primers (Tm) and general suitability were calculated using NetPrimer software (Premier Biosoft International, Palo Alto, CA) and the specificity of the primers was checked with BLASTN (https://blast.ncbi.nlm.nih.gov/Blast.cgi). The PCR mix consisted of 4 µl 2x PCRBIO HS Taq Mix (PCR Biosystem Ltd., London, UK), 0.5 µl EvaGreen (20x, Biotium Inc., Hayward, CA, USA), 1.2 µl gyrA248-L primer, 0.4 µl gyrA248-H primer and 0.4 µl gyrA248-C primer (10 pmol/µl each) for *gyrA*, 0.4 µl parC239-L primer, 0.8 µl parC239-H primer and 0.4 µl parC239-C primer (10 pmol/µl each) for *parC*, 2 µl DNA template and nuclease-free water to a final volume of 12 µl. The melt MAMA assay was optimized for the Bio-Rad CFX96 real-time PCR system using Bio-Rad CFX Maestro software version 1.1 (Bio-Rad Laboratories, Hercules, CA, USA). Thermocycling parameters were 95 °C for 1 min, followed by 40 cycles of 95 °C for 15 sec and 62 °C (for *gyrA*) or 60 °C (for *parC*) for 15 sec. PCR products were subjected to melt analysis using a dissociation protocol consisting of the following steps: 60 °C for 5 sec, followed by a temperature gradient that increases by 0.5 °C up to 95 °C.

## Validation of MAMAs

To evaluate the sensitivity of the assays, 10-fold dilution series were prepared using two of the strains – one inhibited by elevated enrofloxacin MIC and one by low enrofloxacin MIC – in the range of $10^6$ to $10^0$ template copies/µl. The system was also tested on the DNA mixture of high and low MIC strains. This involved combining serotype R (low MIC) and IZSVE/365-2D (high MIC) strains in equal concentrations at the limit of detection, and also with one genotype mixed at the concentration of limit of detection ($10^3$ template copies per reaction) and the other genotype with increasing concentrations ($10^3$–$10^5$ template copies per reaction), and vice versa. The copy number of the template was calculated using the online tool (Thermo Scientific Web Tools – DNA Copy Number Calculator) based on the DNA concentration extracted from the pure *M. iowae* culture, which was measured using a Nanodrop 2000 Spectrophotometer (Thermo Fisher Scientific). The specificity of the assays was evaluated using a diverse range of *Mycoplasma* and non-*Mycoplasma* species isolated from poultry. The following *Mycoplasma* species were subjected to testing: *M. anatis*, *M. anseris*, *M. anserisalpingitidis*, *M. cloacale*, *M. columbinasale*, *M. columborale*, *M. gallinaceum*, *M. gallinarum*, *M. gallisepticum*, *M. gallopavonis*, *M. iners*, *M. meleagridis*, and *M. synoviae*. The following bacterial species were subjected to cross-reaction testing: *Avibacterium*

*paragallinarum*, *Bordetella avium*, *Campylobacter sp.*, *Clostridium sp.*, *Erysipelothrix rhusiopathiae*, *Escherichia coli*, *Gallibacterium anatis*, *Ornithobacterium rhinotracheale*, *Pasteurella multocida*, *Salmonella sp.*, *Staphylococcus sp.* and *Streptococcus sp.*.

## Determination of the optimal concentrations of efflux pump inhibitors

A broth microdilution assay was employed to ascertain the lethal dose of CCCP, OV and RSP (Merck, Rahway, NJ, USA) for the selected 32 *M. iowae* strains showing elevated MIC values to the examined antibiotics (S2 Table). Two-fold dilutions of the freshly prepared EPIs were used in the range of 0.191–97.92 µg/ml for CCCP, 0.0006–0.31 mM for OV, and 0.005–2.56 mg/ml for RSP. CCCP was diluted in 80% ethanol, RSP was diluted in dimethyl sulfoxide (DMSO) (SERVA Electrophoresis GmbH, Heidelberg, Germany) [23]. The preparation of OV was conducted in distilled water, with the pH adjusted to 10.0 using 1N NaOH or 1N HCl. At this point in the process, the solution exhibited a distinctive yellow color. The solution was then heated and boiled until it became colorless (approximately 10 minutes), and all crystals had fully dissolved. All prepared EPIs were stored at −20°C. Standard inocula at $10^4$ color-changing units/ ml (CCU) of *M. iowae* cultures were used in MolliScience General Mycoplasma Liquid Media (MolliScience, Biatorbágy, Hungary). The test was performed on 96-well plates, each strain was examined in duplicates. The plates were incubated at 37°C and checked daily until no further color changes were observed (red to yellow shift). The lowest concentration of the EPI at which no color change was observed was considered to be the lethal dose of the EPI. The optimal concentrations of EPIs required for MIC testing were determined to be at least four-fold below their respective lethal concentrations [24,41].

## Testing MIC with the efflux pump inhibitors

The MIC values of the selected strains were measured for each antibiotic in the presence and absence of EPIs in MolliScience General Mycoplasma Liquid Media (MolliScience). The applied concentrations of the EPIs in the MIC tests were 1.53 µg/ml for CCCP and 0.01 mM for OV. The concentrations of enrofloxacin, tylosin and spiramycin were adjusted within the range of 0.03125–16 µg/ml, while oxytetracycline, tilmicosin and lincomycin were diluted within the range of 0.125–64 µg/ml. Efflux pump inhibitors were freshly prepared before each experiment, and each assay was performed in a 96-well plate. *M. iowae* cultures were diluted at a titer of $10^4$ CCU/ml. Plates were incubated at 37°C and checked daily until no further color change was observed for two days. Each strain was examined in duplicate. Initial MIC values were measured when the growth control (wells containing only the bacterial cultures, without any antibiotics or EPIs) changed color.

The Wilcoxon Signed-Rank test was used to determine whether the MICs were significantly different in the presence or absence of the EPIs using an online tool [36]. For the analyses MIC values equal or lower than the lowest examined concentrations were adjusted to be the lowest examined concentrations, while MIC values higher than the highest examined concentrations were adjusted to be the highest examined concentrations. Furthermore, in order to normalize data distribution (gain linear intervals) the adjusted MIC values from the two-fold dilution series of antimicrobial concentrations were transformed to $\log_2$MIC values [37,38] (S4 Table).

## Identifying putative regulatory regions and possible antibiotic resistance-associated markers of efflux pump coding genes

Putative regulatory sequences were identified in two publicly available whole genome sequences of the *M. iowae* type strain (GenBank accession numbers: LR215023 and CP033512.2.1) using Promotech software version 1.0 [42]. The two sequences exhibited notable differences; thus, the hypothetical putative regulatory sequences occurring in both genomes were used for further analyses. The extracted promoter sequences were mapped to the type strain and the genes associated with efflux pumps were selected. To avoid false positive results, the score threshold was set to be equal or above 0.8

[23]. Therefore, when a sequence was upstream of a gene that codes an efflux pump and scored 0.8 or higher, it was considered a potential regulatory region. The Pribnow boxes and ribosome-binding sites (RBS) of the regulatory sequences were determined manually using the literature [43,44]. The transcription start sites (TSSs) were calculated from the position of the Pribnow boxes. The most common spacer between the TSS and −10 box is 6 nucleotides long, although 5- and 7-nucleotide spacers also exist [43]. The identified sequences were aligned to the whole genome sequences of the 32 *M. iowae* strains (BioProject ID: PRJNA975348) employed for the efflux pump investigation. As in the search for resistance-associated genes, Illumina raw reads of the 32 *M. iowae* strains were first mapped to the *M. iowae* type strain NCTC 10185 (GenBank: LR215023.1) as reference to assemble their whole genome sequences. The SNPs in the efflux pump related regions were identified in the assembled genome sequences in Geneious Prime 2019.2.1 software [35]. The SNPs were subjected to manual review and categorization based on their potential to cause synonymous or non-synonymous SNPs (nsSNPs).

## Results

### Identification of antibiotic resistance-associated genetic markers and development of MAMAs

SNPs were associated with elevated MIC values in ≥80% of the strains in the *gyrA*, *parC* and 23S rRNA genes. In the 23S rRNA gene, only one macrolide and lincomycin resistance-associated mutation was identified at nucleotide position A2059G [*E. coli*]/ A2052T/G/C [*M. iowae*] out of the detected six SNPs (S1 Table). The logistic regression model was statistically significant (chi-square ranged between 46.83–63.53, p < 0.001), and showed strong association between the phenotypes and genotypes (odds ratios ranged between 1.72–3.18, p < 0.001) (S5 Table). The Fisher's exact test also showed strong association between the MIC categories (low or high) and genotypes (wild-type or mutant; p < 0.001, odds ratios ranged between 110.08–182.56) (S5 Table).

A total of two nsSNPs were found in the *gyrA* gene (S1 Table), of which only one SNP showed moderate correlation with MIC values, at nucleotide position C248T [*E. coli*]/ C278T/A [*M. iowae*], resulting in the substitution of serine with phenylalanine or tyrosine. The logistic regression model was statistically significant (chi-square was 44.56, p < 0.001), and MIC values were regarded positive predictors for the mutant genotype (odds ratio 4.51, p < 0.001), however, the baseline odds ratio was very high (35.07) (S5 Table) – although the mutation was present in 92% of the strains with elevated MICs, 60.8% of the strains with low MICs also showed the mutant genotype (Table 1). Association was defined with the Fisher's exact test between the MIC categories (low or high) and genotypes (wild-type or mutant; p = 0.003, odds ratio of 6.09) as well (S5 Table).

Two nsSNPs were identified in the *parC* gene (S1 Table), of which the transversion G239T [*E. coli*]/ G257T [*M. iowae*] leading to the replacement of serine with isoleucine correlated with MIC values (Table 1). The logistic regression model was statistically significant (chi-square of 54.78, p < 0.001), and showed strong association between elevated enrofloxacin MIC values and genotypes (odds ratio of 4.03, p < 0.001). The Fisher's exact test also showed strong association (p < 0.001, odds ratio of 33.55) (S5 Table).

No correlations were found among elevated MIC values and nucleotide variants at positions previously described to be associated with antimicrobial resistance in the other examined regions of the *M. iowae* strains (*gyrB*, *parE* genes, 50S ribosomal proteins L4 and L22).

The mutation associated with antibiotic resistance in the 23S rRNA gene proved to be unsuitable for MAMA design, as the transition (A/T) and transversions (A/G and A/C) were all detected at this position (2059 [*E. coli*]/ 2052 [*M. iowae*]). The most frequently observed mutation was A/G transversion, presented in 63.3% (n = 50/79) of the strains showing mutation at this position and elevated MICs to macrolides and lincomycin. For fluoroquinolones, the developed MAMAs clearly differentiated the *M. iowae* strains' genotypes associated with low (L) or high (H) MICs of enrofloxacin. In the *gyrA* gene the transversion C/T (n = 21/23) was more frequently detected than the C/A mutation (n = 2/23). The MAMA targeting the mutation C248T [*E. coli*]/ C278T [*M. iowae*] resulted genotype L for strains showing

**Table 1. Potentially resistance-related mutations identified in *M. iowae* strains.**

| Antibiotics | Genes | Position of mutation[a] | Strains possessing the mutation and showing lower MIC values relative to all strains with low MICs[b] | | Strains possessing the mutation and showing higher MIC values relative to all strains with high MICs[c] | |
|---|---|---|---|---|---|---|
| | | | No. | % | No. | % |
| Enrofloxacin | *gyrA* | C248T/A | 45/74 | 60.8 | 23/25 | 92.0 |
| | *parC* | G239T | 7/74 | 9.5 | 20/25 | 80.0 |
| | *gyrA* & *parC* | both C248T/A & G239T | 6/74 | 8.1 | 19/25 | 76.0 |
| Erythromycin | 23S rRNA | A2059G/T/C | 1/16 | 6.3 | 79/83 | 95.2 |
| Spiramycin | | | 1/15 | 6.7 | 79/84 | 94.1 |
| Tilmicosin | | | 1/16 | 6.3 | 79/83 | 95.2 |
| Tylosin | | | 1/14 | 7.1 | 79/85 | 92.9 |
| Lincomycin | 23S rRNA | A2059G/T/C | 1/15 | 6.7 | 79/84 | 94.1 |

Abbreviations: No.= number of strains, % = percentage of strains

[a]according to *E. coli* numbering

[b]in case of fluoroquinolones: strains with MIC values of ≤0.5 μg/ml for enrofloxacin (n = 74); in case of macrolides: strains with MIC values of ≤1 μg/ml for erythromycin (n = 16), ≤ 2 μg/ml for spiramycin (n = 15), ≤ 8 μg/ml for tilmicosin (n = 16), ≤ 1 μg/ml for tylosin (n = 14); in case of lincomycin: strains with MIC values of ≤1 μg/ml (n = 15).

[c]in case of fluoroquinolones: strains with MIC values of >0.5 μg/ml for enrofloxacin (n = 25); in case of macrolides: strains with MIC values of >1 μg/ml for erythromycin (n = 83), > 2 μg/ml for spiramycin (n = 84), > 8 μg/ml for tilmicosin (n = 83), > 1 μg/ml for tylosin (n = 85); in case of lincomycin: strains with MIC values of >1 μg/ml (n = 84)

cytosine or genotype H for strains showing thymine or adenine. The primer specific for genotype H has adenine at the SNP specific position at the 3' end, thus it showed lower sensitivity for strains with C/A transversion (ct values of the $10^3$ dilutions of the two strains – which possessed the C/A mutation – were compared in the *gyrA* and *parC* specific MAMAs: B2/90 ct 34 and 32, respectively; B67/01 ct 32 and 30, respectively). The MAMA targeting the mutation G239T [*E. coli*]/ G257T [*M. iowae*] in the *parC* gene resulted genotype L for guanine or genotype H for thymine (Table 2). In the *gyrA* system, the average melting temperatures (Tm) for samples with genotype H were 81.79 °C ± 0.62 and 77 °C ± 0.63 with samples of genotype L. In the *parC* system the average Tm values were 79.7 ± 0.52 and 76.6 ± 0.63 (Fig 1) for genotypes H and L, respectively.

**Table 2. Primer sequences and parameters of the designed MAMAs.**

| Targeted gene and SNP | Primer name* | Primer sequence (5'-3')** | Geno-type* | Amplicon size (bp) | Average melting temperature of the amplicons (°C) |
|---|---|---|---|---|---|
| *gyrA* gene C248T | gyrA248-L | CCATTCTAACCATAGTTTCATAAACTGT**AG** | L | 83 | 77 |
| | gyrA248-H | ggggcggggcggggcCCATTCTAACCATAGTTTCATAAACTG**GAA** | H | 101 | 81.8 |
| | gyrA248-C | GTCAGCTAGACTTGTTGGGGAAG | | | |
| *parC* gene G239T | parC239-L | GTAAATATCACCCACATGGTGA**GAG** | L | 118 | 76.6 |
| | parC239-H | ggggcggggcggggcTAAATATCACCCACATGGTGA**CAT** | H | 136 | 79.6 |
| | parC239-C | CCATCAAGAGAACCTTTGTTACCT | | | |

*Abbreviations: L = genotype corresponding to low MIC values, H = genotype corresponding to high MIC values, C = consensus

**Genotype specific and destabilizer nucleotides are in bold; GC-clamp is in lowercase

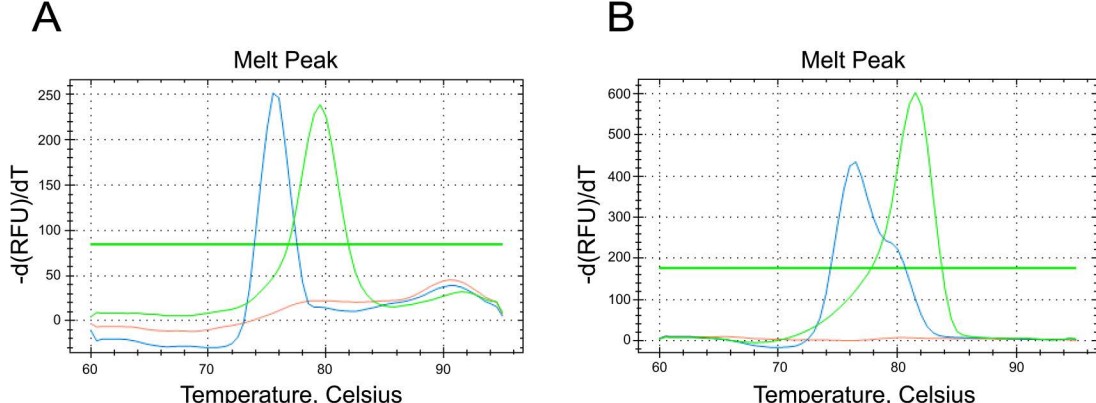

**Fig 1. Mismatch amplification mutation assays targeting the SNPs in the *parC* and *gyrA* genes.** A – Melting curves of *M. iowae* reference strain serotype R (blue, genotype **L**), strain IZSVE/365-2D (green, genotype H) and negative control (orange) in *parC* system. **B** – Melting curves of *M. iowae* reference strain serotype R (blue, genotype **L**), strain IZSVE/365-2D (green, genotype H) and negative control (orange) in *gyrA* system.

## Validation of MAMAs

The sensitivity of the assay was $10^3$ template copies/reaction in both the *gyrA* and *parC* specific systems, for both genotypes, and no cross-reactions were detected in any examined cases. Samples of mixed genotypes showed either a single melt peak (dominant genotype) or two distinct peaks corresponding to both genotypes, depending on the relative copy number (Table 3).

## Determination of the optimal concentrations of efflux pump inhibitors

The growth of the strains was not affected by RSP, color change was detected at the same time as the growth control even at the highest tested concentration (2.56 mg/ml); therefore, this EPI was excluded from further examinations. For CCCP, the lethal dose was determined to be 6.12 μg/ml. The lethal dose was 6.12 μg/ml for 47% (n = 15) of the strains, while 12.24 μg/ml for another 47% (n = 15), and 24.48 μg/ml for the remaining 6% (n = 2) of the strains. The optimal dose (four-fold below the minimal lethal concentration) was determined to be 1.53 μg/ml, and examinations were performed at a double concentration (3.06 μg/ml) of CCCP also. For OV the minimal lethal dose was 0.03875 mM and the optimal concentration used for the MIC testing was 0.00969 mM. The majority of the strains (88%, n = 28) exhibited a lethal dose of 0.03875 mM, while a smaller proportion demonstrated a lethal dose of 0.0775 mM (9%, n = 3), and a minority (3%, n = 1) exhibited a lethal dose of 0.155 mM.

## Testing MICs with the efflux pump inhibitors

Six strains with slightly, moderately or highly elevated MICs were selected for the examination of efflux activity for each of the six antibiotics based on their previously published MIC values determined in Oxoid Mycoplasma broth medium (pH

**Table 3. Melting temperatures (°C) of mixed genotypes in the MAMAs targeting SNPs in the *parC* and *gyrA* genes.**

| Targeted gene and SNP | Genotype L | Genotype H | Ratio of genotype L and genotype H | | | | |
|---|---|---|---|---|---|---|---|
| | | | $10^5{:}10^3$ | $10^4{:}10^3$ | $10^3{:}10^3$ | $10^3{:}10^4$ | $10^3{:}10^5$ |
| *parC* gene G239T | 76.6 | 79.6 | 76 | 76 | 76 | 79 | 79 |
| *gyrA* gene C248T | 77 | 81.8 | 77 & 81.8 | 77 & 81.8 | 77 & 81.8 | 77 & 81.8 | 80 |

7.8) (Thermo Fisher Scientific Inc., Oxoid Inc.) supplemented with 0.5% (w/v) sodium pyruvate, 0.5% (w/v) glucose, 0.15% (w/v) L-arginine hydrochloride and 0.005% (w/v) phenol red [33]. Two-fold differences in MICs can occur due to measurement inconsistencies; thus, these were not considered as changes in the MICs. Higher than one dilution step (two-fold concentration) differences were observed between MIC values obtained in Oxoid medium and in MolliScience General Mycoplasma Liquid Media used in the current study for enrofloxacin (n = 5/6 strains), oxytetracycline (n = 3/6 strains), tilmicosin (n = 1/6 strain) and tylosin (n = 1/6 strain). In this study, the comparisons of enrofloxacin and lincomycin MIC values in MolliScience General Mycoplasma Liquid Media in the presence or absence of OV revealed at least four-fold lower MIC values due to OV for two strains in each case. Regarding oxytetracycline, tilmicosin and tylosin, the MIC values for five strains were at least four-fold lower in the presence of OV and in case of spiramycin, the addition of OV showed at least six-fold decrease in MIC values for half of the examined strains (S2 Table, Fig 2). A statistically significant difference was identified between all the MICs in the presence and absence of OV (p < 0.001, S6 Table). The MIC values in MolliScience General Mycoplasma Liquid Media in the presence and absence of CCCP (both at 1.53 µg/ml and 3.06 µg/ml concentrations) were the same, thus the comparison revealed no effect of this EPI on the examined *M. iowae* strains.

**Identifying putative regulatory regions and possible antibiotic resistance-associated markers in efflux pump genes**

A total of 12 efflux pump-related genes were examined, and promoter regions were identified in five of them (S2 Table). These included the cobalt transporter ATP-binding subunit, the ABC transporter ATP-binding protein, the uncharacterized ABC transporter permease MG468 homologue, the ABC-type multidrug/ protein/ lipid transport system ATPase component, and the MATE efflux family protein genes. The most conserved elements were identified as the Pribnow boxes (-10 boxes) and the initiator nucleotide (A or G). The Pribnow boxes and the RBSs were identified in all five promoters, and the transcription start sites (TSSs) were also calculated (Table 4). There were no pronounced −35 boxes in the promoter region.

Non-synonymous SNPs were identified in the following efflux pump-related genes: ABC transporter ATP-binding protein, ATP-binding cassette domain-containing protein, energy-coupling factor transporter transmembrane protein and MATE family efflux transporter CDS (S2 Table). No correlation was identified between the nsSNPs in the strains and the effect of OV. On the other hand, strains from the previously determined MLST clusters A and B [34] contained the same nsSNPs in the ABC transporter ATP-binding protein genes, and a similar pattern was observed in the energy-coupling factor transporter transmembrane protein and ATP-binding cassette domain-containing protein genes. Likewise, MLST cluster D strain specific nsSNPs were identified in the ABC transporter ATP-binding protein- *lol*D2, and nsSNPs specific for strains from the MLST cluster C were detected in the uncharacterized ABC transporter permease MG468 homolog (S2 Table).

## Discussion

*M. iowae* infection is economically significant in the turkey industry, and it has been regarded as the most resistant avian pathogen *Mycoplasma* species to antibiotics in previous publications [1,7,45]. The higher mutation frequency in *M. iowae* compared to other mycoplasmas may enable the strains to rapidly and efficiently develop resistance to antibiotics [7]. Furthermore, its unique ecological niche (adaptation to the intestinal environment) and its relative neglect by pathologists and practitioners may constitute key predisposing factors in the notable resistance to antimicrobial compounds. *M. iowae* may persist in the turkey's intestinal tract for the entire commercial lifespan [46], being exposed to a wider spectrum of antimicrobials substantially different from the ones administered against the respiratory pathogen mycoplasmas. The emergence of resistance in avian mycoplasmas demands a rigorous and prudent approach to antibiotic use [11]. To our knowledge, this study is the first to investigate various resistance mechanisms in *M. iowae* with the aim to support targeted antibiotic treatment by providing basic information about efflux pump activity in this species and by identifying antibiotic resistance related mutations and developing molecular tools for their rapid detection.

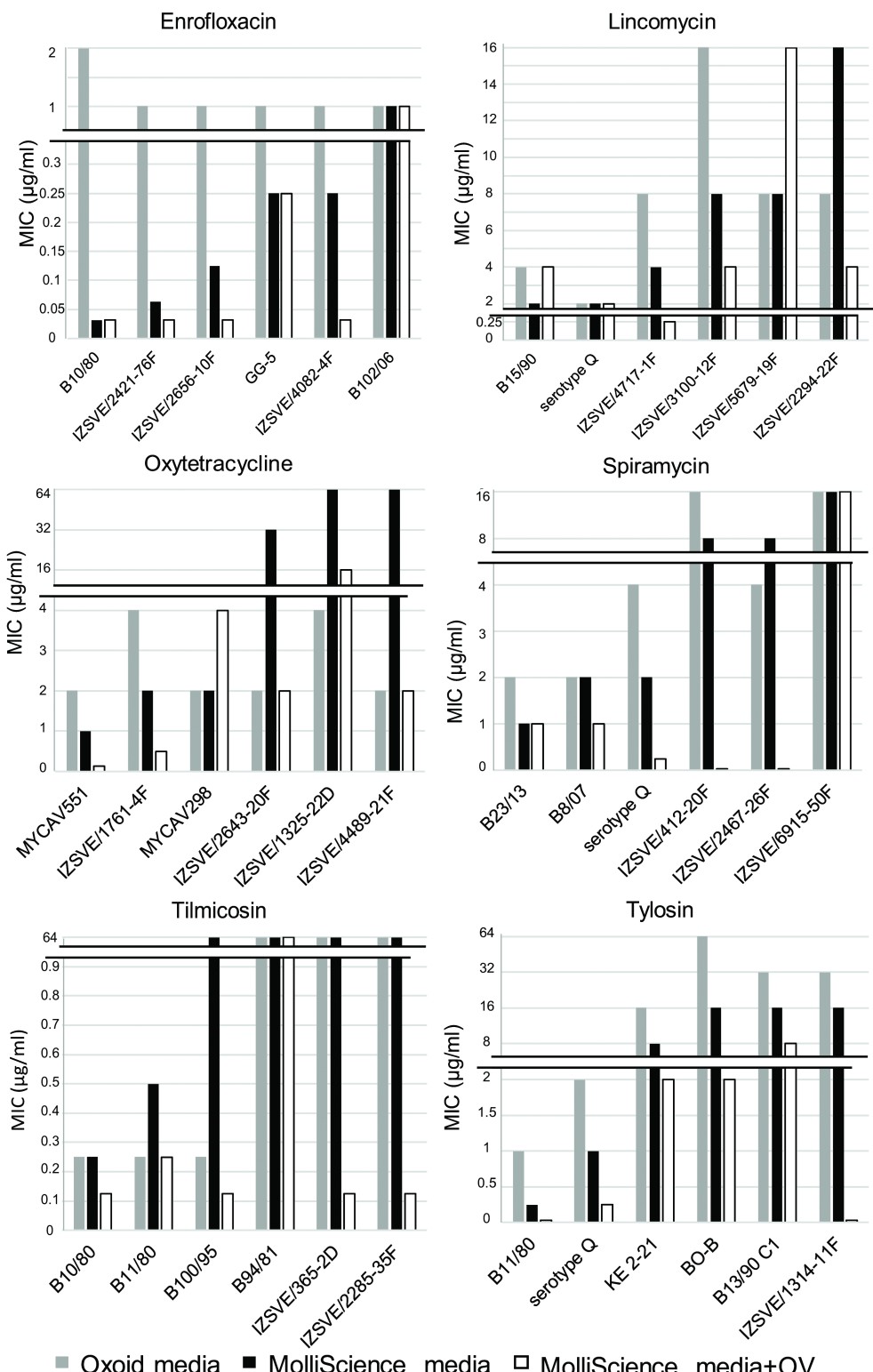

**Fig 2. The effect of sodium orthovanadate on MIC values in *Mycoplasma iowae* strains.** The MIC values of enrofloxacin, lincomycin, oxytetracycline, spiramycin, tilmicosin and tylosin against six *M. iowae* strains are shown in Oxoid Mycoplasma broth medium (grey columns, "Oxoid media"), in MolliScience General Mycoplasma Liquid medium (black columns, "MolliScience media"), and in MolliScience General Mycoplasma Liquid medium supplemented with sodium orthovanadate (white columns, "MolliScience+OV"). The y axis is split on all diagrams to better visualize MIC values of great difference.

**Table 4. Putative regulatory sequences of the efflux pump-related genes.** The potential Pribnow boxes (lowercase), ribosome-binding sites (underlined), transcription start site (black bold) and the start codon (italics) are highlighted.

| Gene name | Promoter sequence with the motifs (5'-3') |
|---|---|
| MATE efflux family protein CDS | TGTTATtataatAACATC**A**CAGATTTATTTAATGC-CA<u>GGT</u>CTTCCTAAAGTACCTAAAGCAAAGAGT-TAATT*ATG* |
| ABC-type multidrug/ protein/lipid transport system ATPase component-mldB1 | TTTTAAATTAAATATTGTATtataatTCATGT**G**AAT-GTTTTTTAATAATCAGTTACGAAAGCG<u>GGT</u>TATA-ATATA*ATG* |
| cobalt transporter ATP-binding subunit-cbiO4 | AATTTCATTCTTTTTTATTTtataatTTAtataatTTTA-A**G**AATT<u>GGG</u>ATTTATTT*ATG* |
| ABC transporter ATP-binding protein- lolD2 | CTATTTTTCTTtataatTtataatTAATA**A**GAAATAAAC-CA<u>GGT</u>AAGATT*ATG* |
| Uncharacterized ABC transporter permease MG468 homolog | TTACTTATAAAGTTTTTTTATAAtataatAAATA**A**TTA-AAAAAACACAACAAAAAAATAAA<u>GGT</u>AATT*ATG* |

Genetic markers associated with elevated MIC values in fluoroquinolones, macrolides and lincosamides were evaluated based on the antibiotic susceptibility profiles and whole genome sequences of a diverse selection of 99 *M. iowae* strains. Macrolides inhibit bacterial protein synthesis by binding to the 50S subunit of the 23S rRNA, which overlaps with the binding site of lincosamides; thus, resistance of these agents is linked [47]. A SNP correlating with elevated MICs for macrolides and lincosamides was identified in the 23S rRNA gene at 2059 [*E. coli*]/ 2052 [*M. iowae*] position in the examined *M. iowae* strains in this study. While the A2059G substitution has most commonly been demonstrated to be associated with reduced susceptibility in other studies [11], all four nucleotides were observed at nucleotide position 2059 [*E. coli*]/ 2052 [*M. iowae*] in *M. iowae* – a phenomenon which has also been described in other mycoplasmas [9,48]. Unfortunately, the high nucleotide variability impeded the development of a MAMA for their differentiation in *M. iowae*. Furthermore, four strains which showed decreased susceptibility to macrolides and lincomycin (strains B15/90, IZSVE/3336-1F, IZSVE/365-2D and IZSVE/2285-35F) revealed the wild-type genotype, while one strain showing high susceptibility to these antimicrobials possessed the mutant genotype (strain IZSVE/6456-2F, isolated from a duck) (S1 Table). These unexplained resistant/susceptible phenotypes indicate the significant contribution of other mechanisms in the development of macrolide and lincomycin resistance of *M. iowae*.

Fluoroquinolones are synthetic antibacterial drugs that act on both DNA gyrase and topoisomerase IV [49]. Mutations in the quinolone resistance determining region (QRDR) of the pathogen result in the inhibition of fluoroquinolone binding to both DNA and enzyme complexes, which is the primary cause of fluoroquinolone resistance [49]. Resistance to fluoroquinolones in mycoplasmas is frequently the result of target gene mutations within the *gyrA* and *parC* genes [10,50]. In *M. bovis* it has been described that a single change in *gyrA* alone is sufficient to achieve moderate resistance to fluoroquinolones, but a simultaneous modification in *parC* is necessary to achieve high MICs [16,51,52]. In the present study, moderate association was defined between resistant phenotypes and the C248T [*E. coli*]/ C278T/A [*M. iowae*] mutation in *gyrA* gene, with slight elevation in MIC values when the identified mutation was present alone (36 strains showed MICs between 0.25–0.5 µg/ml from the 43 strains possessing the SNP in the *gyrA* gene only, S1 Table). It is noteworthy, that this substitution was presented in more than half of the examined *M. iowae* strains (68.7%, n = 68/99). The presence of the G239T [*E. coli*]/ G257T [*M. iowae*] substitution in the *parC* gene alone was observed only in two strains from the examined 99 *M. iowae* strains (with MICs 0.5 and 4 µg/ml), while the detection rate of this mutation was as low as 27.3% (n = 27/99) in all the examined strains. The presence of mutations in both *gyrA* and *parC* genes was associated with an increase in MIC values above 0.5 µg/ml (19 strains showed MIC > 0.5 µg/ml from the 25 strains possessing both resistance-associated SNPs, Table 1), consistent with previous findings [16,51,52]. Nevertheless, 74.1% (n = 20/27) of

the strains possessing the G239T [*E. coli*]/ G257T [*M. iowae*] substitution in the *parC* gene, and 33.8% (n = 23/68) of the strains with the C248T [*E. coli*]/ C278T/A [*M. iowae*] substitution in the *gyrA* gene were inhibited only by elevated MICs of enrofloxacin (>0.5 µg/ml). The described association between elevated MIC values and the mutant *parC* genotype indicates that the identified SNP is a reliable target for molecular assays, while the additional differentiation of the SNP C248T [*E. coli*]/ C278T [*M. iowae*] in the *gyrA* gene may support the preliminary detection of resistant phenotypes. Therefore, the developed MAMAs, targeting these two resistance markers, offer a time-efficient and cost-effective detection system that can also be used on clinical samples to predict *M. iowae* strains with elevated MIC values for fluoroquinolones, which may facilitate quick decision making for targeted antibiotic therapy during outbreaks. It should be considered though, that the application of molecular assays for the detection of antimicrobial resistance has several limitations (e.g., only known mutations are targeted, resistance clones may vary by geographical regions, inconsistency of assay validations and sensitivity) [39]. Moreover, as for the conventional, isolate-based microdilution tests, clinical samples with strains of mixed genotypes represent another challenge in molecular diagnostics as well. For instance, the developed MAMA targeting the mutation in the *parC* gene would identify only the wild-type when the ratio of the wild-type and mutant strains is equal in the sample. Also, given the detected low number of SNPs correlating with elevated MIC values in *M. iowae*, the significance of other antibiotic resistance methods is suggested to be higher in this pathogen and warrants future examinations.

Efficient efflux pump mechanisms have been previously identified in other mycoplasmas, with strains exhibiting slightly elevated MIC values [17,23]. In the current study, the selection of *M. iowae* strains (inhibited by slightly or moderately elevated MIC values) for the examinations was made based on the evidence that efflux pumps have an impact at the early stage of the resistance development [17]. Due to the rapidly developing resistance in *M. iowae* the number of selectable strains was restricted in the current dataset; therefore, in certain cases (especially for macrolides), multiple strains with originally relatively high MIC values (upper values of the examined concentration range) were involved in the tests. The initial MIC values for enrofloxacin and oxytetracycline were notably different from those obtained in our previous study [33], although *M. iowae* strains reached comparable CCU/ml in the two media (the strains reached $10^7$–$10^9$ CCU/ml in both media used in the two studies), which is – in the absence of standardizations – the only criterion for media in MIC testing of veterinary mycoplasmas [8,39,53]. Out of the three inhibitors tested, the use of CCCP did not result in any difference in the MICs, while reserpine did not even affect the growth of the strains and examinations were discontinued. However, as reserpine did not change viability of *Mycoplasma pneumoniae* cells earlier either, but its effect on azithromycin MIC values was detectable when applied at 20 µg/ml concentration (four-fold changes in two cases, and two-fold changes in 25 cases out of the 30 examined resistant isolates) [54], it cannot be excluded that if applied in the appropriate concentration reserpine would have modulated efflux pump activity in the examined *M. iowae* isolates. In the presence of OV, the MIC values against the *M. iowae* isolates were significantly lower than in the absence of this EPI in the tests with four antibiotics (oxytetracycline, spiramycin, tilmicosin and tylosin; Fig 2). Interestingly, the greatest effect of OV was observed for macrolides with strains showing originally relatively high MIC values, suggesting more advanced resistance mechanisms. By coincidence, two isolates (IZSVE/365-2D and IZSVE/2285-35F) with highly decreased susceptibility to macrolides and lacking the mutation at the position 2059 [*E. coli*]/ 2052 [*M. iowae*] in the 23S coding region were also included in the OV tests. In these cases, significant decrease of tilmicosin MIC values was detected when OV was applied. For lincomycin and enrofloxacin, although less than half of the strains exhibited decrease in the MICs in the presence of OV, those differences were at least four-fold concentrations. The findings indicate that OV influences the antimicrobial susceptibility of *M. iowae*, but the involvement of efflux pump mechanisms in the resistance development is yet to be confirmed by further transcriptional assays or validations at protein-level.

The number of strains involved in the examination of efflux pump associated gene sequences did not permit in-depth analyses regarding correlation between SNPs and the reduced MICs observed in the presence of OV (S2 Table). The computational prediction of promoters also represents a significant challenge in the field of characterization of bacterial transcriptional units. The majority of promoter-finding algorithms rely on the sequence motifs recognized by sigma factors,

which guide the RNA polymerase complex to TSSs [55]. In many bacteria, the housekeeping transcription factor sigma 70 binds to two regions upstream of the TSS. The first of these is located 10 bp upstream of the TSS (Pribnow box) and bears the consensus motif TANAAT (N can be any base) [56]. The second is around 35 bp upstream of the TSS and bears the motif TTGACA (−35 box). The spacer between these two boxes may extend from 15 to 21 bases in length [55,57]. The Pribnow boxes with (TATAAT), RBSs (GGK), and TSSs were identified in the efflux gene promoters described in the present study. However, no −35 boxes were found, presumably due to the less conserved nature of this sequence in mycoplasmas. In AT-rich organisms, such as mycoplasmas, it is suggested that the AT-rich regions serve as regulatory up elements, potentially decreasing the necessity of −35 boxes [58]. Several nsSNPs were identified in promoter-associated genes (S2 Table), although an exact correlation between these and the observed reduction in MIC values in the presence of the EPI could not be established. Nevertheless, these SNPs may influence the function of the efflux pumps, as in the case of *Salmonella enterica* serovar Typhimurium. In this example, an insertion in the promoter region of the genes encoding the AcrEF pump has been shown to enhance the transcription of the corresponding operon, which leads to an increase in resistance to fluoroquinolones. However, it is notable that the majority of mutations affecting antibiotic export by efflux systems occur in genes encoding proteins that perform regulatory functions [59]. The current study presents the first inhibitor-based investigation of the involvement of efflux pump mechanism in antimicrobial resistance and its underlying genetic background in *M. iowae*. The identification of promoter sequences in the present study represents an important step forward to a deeper insight into the mechanisms of gene regulation. Yet, there is a clear need for further research to better understand the regulation in transcriptional initiation in *Mycoplasma*.

The study identified known nucleotide positions for target modification, and described the effect of an EPI on MIC against *M. iowae* suggesting the possibility of efflux pump activity in the development of antimicrobial resistance, but the strains which revealed no changes from these aspects may indicate the presence of alternative targets or mechanisms. Biofilm formation of a limited number of *M. iowae* strains was examined in a previous experiment [60], which could not confirm this capability of the pathogen either. Therefore, a deeper understanding of the efflux pump mechanisms may facilitate the advancement of supplementary techniques to antibiotic therapy and a more comprehensive understanding of the molecular basis of resistance mechanisms in *M. iowae* helps the development of new methods for the detection of antibiotic susceptibility. Furthermore, the potential acquisition of resistance determinants characteristic of enteric bacteria should be taken into account in future examinations. Identification of novel resistance markers and mechanisms will enable the early prediction of antibiotic efficacy and the mitigation of public health concerns related to bacterial resistance.

## Supporting information

**S1 Table. Details of the 99 *Mycoplasma iowae* strains and the nsSPNPs in the *gyrA, parC* and 23S rRNA genes.**
(XLSX)

**S2 Table. nsSNPs in potential efflux pump related genes and the initial MIC values of the six antibiotics, in the absence and presence of sodium orthovanadate.**
(XLSX)

**S3 Table. Data used for logistic regression and Fisher's exact test to analyze associations among MIC values and genotypes in *Mycoplasma iowae*.**
(XLSX)

**S4 Table. Data used for Wilcoxon Signed-Rank test to analyze the effect of sodium orthovanadate on MIC values against *Mycoplasma iowae* strains.**
(XLSX)

 

**S5 Table. Results of the logistic regression analysis and Fisher's exact test regarding associations among MIC values and genotypes in *Mycoplasma iowae*.**
(XLSX)

**S6 Table. Results of the Wilcoxon Signed-Rank test regarding the effect of sodium orthovanadate on MIC values against *Mycoplasma iowae* strains.**
(XLSX)

## Author contributions

**Conceptualization:** Enikő Wehmann, Miklós Gyuranecz, Zsuzsa Kreizinger.

**Data curation:** Áron Botond Kovács, Zsuzsa Kreizinger.

**Formal analysis:** Dominika Buni, Áron Botond Kovács, Enikő Wehmann.

**Funding acquisition:** Miklós Gyuranecz, Zsuzsa Kreizinger.

**Investigation:** Dominika Buni, Eszter Zsófia Nagy.

**Methodology:** Dominika Buni, Enikő Wehmann, Krisztián Bányai, Eszter Zsófia Nagy, Zsuzsa Kreizinger.

**Project administration:** Dominika Buni, Dénes Grózner.

**Resources:** Dénes Grózner, Krisztián Bányai, Janet Bradbury, Marco Bottinelli, Elisabetta Stefani, Salvatore Catania, Inna Lysnyansky, László Kovács.

**Supervision:** Enikő Wehmann, Krisztián Bányai, Eszter Zsófia Nagy, Miklós Gyuranecz, Zsuzsa Kreizinger.

**Validation:** Áron Botond Kovács, Dénes Grózner, Zsuzsa Kreizinger.

**Visualization:** Dominika Buni.

**Writing – original draft:** Dominika Buni.

**Writing – review & editing:** Áron Botond Kovács, Enikő Wehmann, Dénes Grózner, Krisztián Bányai, Eszter Zsófia Nagy, Janet Bradbury, Marco Bottinelli, Elisabetta Stefani, Salvatore Catania, Inna Lysnyansky, László Kovács, Miklós Gyuranecz, Zsuzsa Kreizinger.

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
