## [Decision Letter · Decision Letter 0]

2 Jan 2026

Dear Dr. Kreizinger,

Thank you for submitting your manuscript to PLOS ONE. After careful consideration, we feel that it has merit but does not fully meet PLOS ONE’s publication criteria as it currently stands. Therefore, we invite you to submit a revised version of the manuscript that addresses the points raised during the review process.

We look forward to receiving your revised manuscript.

Kind regards,

Chih-Horng Kuo, Ph.D.

Academic Editor

PLOS One

Journal Requirements:

This work was supported by the FK21 (137809) grant of the National Research, Development and Innovation Office, Hungary; the SA-27/2021 grant of the Eötvös Loránd Research Network; the Projects no. RRF-2.3.1-21-2022-00001 and RRF-2.3.1-21-2022-00006 which have been implemented with the support provided by the Recovery and Resilience Facility (RRF), financed under the National Recovery Fund budget estimate, RRF-2.3.1-21 funding scheme; and the support provided by the Ministry of Innovation and Technology of Hungary (legal successor: Ministry of Culture and Innovation of Hungary) from the National Research, Development and Innovation Fund, financed under the TKP2021-EGA-01 funding scheme of the National Research, Development and Innovation Office. The funders had no role in study design, data collection and interpretation, or the decision to submit the work for publication.

This work was supported by the FK21 (137809) grant of the National Research, Development and Innovation Office, Hungary; the SA-27/2021 grant of the Eötvös Loránd Research Network; the Projects no. RRF-2.3.1-21-2022-00001 and RRF-2.3.1-21-2022-00006 which have been implemented with the support provided by the Recovery and Resilience Facility (RRF), financed under the National Recovery Fund budget estimate, RRF-2.3.1-21 funding scheme; and the support provided by the Ministry of Innovation and Technology of Hungary (legal successor: Ministry of Culture and Innovation of Hungary) from the National Research, Development and Innovation Fund, financed under the TKP2021-EGA-01 funding scheme of the National Research, Development and Innovation Office. The funders had no role in study design, data collection and interpretation, or the decision to submit the work for publication.

This work was supported by the FK21 (137809) grant of the National Research, Development and Innovation Office, Hungary; the SA-27/2021 grant of the Eötvös Loránd Research Network; the Projects no. RRF-2.3.1-21-2022-00001 and RRF-2.3.1-21-2022-00006 which have been implemented with the support provided by the Recovery and Resilience Facility (RRF), financed under the National Recovery Fund budget estimate, RRF-2.3.1-21 funding scheme; and the support provided by the Ministry of Innovation and Technology of Hungary (legal successor: Ministry of Culture and Innovation of Hungary) from the National Research, Development and Innovation Fund, financed under the TKP2021-EGA-01 funding scheme of the National Research, Development and Innovation Office. The funders had no role in study design, data collection and interpretation, or the decision to submit the work for publication.

Additional Editor Comments:

Based on the expert reviewer’s detailed evaluation and my editorial assessment, I invite you to submit a major revision of this manuscript. The reviewer raised substantive concerns regarding the completeness and interpretation of resistance mechanisms, the lack of statistical support for genotype–phenotype associations, and the strength of conclusions drawn about efflux pump activity. As the handling editor with expertise in Mycoplasma genomics, I agree with these concerns and consider them important to address before further consideration.

Due to the timing of the review process, I am proceeding with a decision based on the review comments received to date. Additional expert input may be sought after revision, depending on the extent of the changes.

In particular, please ensure that the revised manuscript:

- Clarifies which resistance mechanisms were examined and explicitly discusses unexplained resistant phenotypes as limitations;

- Provides appropriate statistical support for SNP–MIC associations or clearly frames these analyses as descriptive;

- Moderates and qualifies conclusions regarding efflux pump involvement, taking into account the limitations of inhibitor-based assays;

- Aligns claims of novelty and diagnostic applicability with the scope and strength of the data.

The reviewer’s additional technical and minor comments should also be addressed in your revision.

Reviewers' comments:

Reviewer's Responses to Questions

**Comments to the Author**

1. Is the manuscript technically sound, and do the data support the conclusions?

Reviewer #1: Partly

2. Has the statistical analysis been performed appropriately and rigorously?

Reviewer #1: No

3. Have the authors made all data underlying the findings in their manuscript fully available?

Reviewer #1: Yes

4. Is the manuscript presented in an intelligible fashion and written in standard English?

Reviewer #1: Yes

Reviewer #1: This manuscript investigated mechanisms of antimicrobial resistance in Mycoplasma iowae using genomic and phenotypic approaches and developed molecular diagnostic tools for resistance detection. The authors analyzed genome sequences of 99 strains to identify SNPs associated with elevated MICs to fluoroquinolones, tetracyclines and macrolides; developed mismatch amplification mutation assays (MAMAs) for rapid detection of fluoroquinolone resistance markers; and evaluated the role of efflux pumps using efflux inhibitor assay, as well as explored potential regulatory regions of efflux pump genes using in silico analysis. This work provides molecular tools for fluoroquinolone resistance detection in M. iowae and demonstrates the involvement of active efflux pump in antimicrobial susceptibility in M. iowae, which fills an important knowledge gap.

Below are my comments/suggestions:

Major Concerns:

1. Incomplete results of resistance mechanisms

The genetic mechanisms associated with elevated MICs to tetracyclines are not shown, although searching for SNPs in 16S rRNA gene was mentioned in Materials and Methods. It would also be interesting to check other tetracycline resistance mechanisms, such as mobile genetic elements carrying tetracycline resistance genes (e.g. tetM, tetO) and mutations in ribosomal proteins in the 99 isolates. For macrolide resistance, only SNPs in 23S rRNA are presented while no results for ribosomal proteins L22 and L23 (should be L4?). There are 4 isolates with high MICs for macrolides/ lincomycin without alteration in position 2059 in 23S rRNA. This needs to be investigated.

2. Lack of statistical support of genotype–phenotype associations

The manuscript lacks statistical analysis (e.g., Fisher’s exact test or logistic regression) comparing MIC distributions between wild-type and mutant genotypes for gyrA, parC, and 23S rRNA.

3. Limited efflux pump functional validation

Efflux pump activity validation is based on the MIC reductions by inhibitors, particularly with orthovanadate. Reserpine was excluded from the evaluation due to no effects on growth with high dose. However, experiences from other organisms, including Mycoplasma pneumoniae, even with a very high lethal dose, low level of reserpine still could reduce the MICs of certain drugs like macrolides. It would be a good completion to have this done in M. iowae. Another concern is that there is no transcriptional (e.g., RT-qPCR or RNA-seq) or protein-level validation.

Minor Comments:

1. Abstract is long. Consider reducing the length and clarifying that MAMAs were developed only for fluoroquinolone resistance-associated SNPs.

2. Line 245: “and were aligned” should be deleted.

3. Clarify that the MAMA primers are allelic specific. The text contents in line 279-280 do not match the gyrA primers in Table 2: primer gyrA248-L is for genotype L (wild type, cytosine at 248, not adenine) and gyrA248-H is only for mutant strain carrying a thymine at 248 (not cytosine or adenine). There is no allelic specific primer for 248A.

4. According to Table 3, parC genotype L and H in a 1:1 mixed ratio (103:103) would be interpreted as L genotype. This will lead to false negative results for H genotype in application. This limitation should be discussed.

5. In Figure 2, include all six antimicrobials to make a whole view.

6. In Table S2, the 12 efflux pump gene names should be shown with their positions.

.

Reviewer #1: No

---

## [Author Response · Author response to Decision Letter 1]

27 Feb 2026

Zsuzsa Kreizinger

HUN-REN Veterinary Medical Research Institute

Hungária krt. 21.

H-1143 Budapest, Hungary

Phone: +36 1 467 4060 Fax: +36 1 467 4076

E-mail: kreizinger.zsuzsa@vmri.hun-ren.hu

Chih-Horng Kuo

Academic Editor

PLOS One

27th February, 2026

Dear Chih-Horng Kuo,

We thank You and the Reviewer for Your e-mail on 3rd of January, 2026 and for Your careful evaluation of our manuscript” Identification and detection of genetic markers associated with antimicrobial susceptibility and evaluation of efflux pump mechanisms in Mycoplasma iowae” and for the constructive comments provided. The suggestions have significantly improved the clarity, and overall quality of the manuscript.

I. Response to Journal Requirements

1. Compliance with style requirements and file naming

We have thoroughly reviewed the manuscript to ensure full compliance with the journal’s style and formatting requirements, including correct file naming conventions. Hopefully, all structural, formatting, and stylistic requirements have now been met.

2. Code sharing requirements

We fully agree with the journal’s policy regarding code transparency and reproducibility. In the revised version of the manuscript programs (Geneious, Promotech) were used according to the best practice that their developers recommend for sequence analyses, and online tools (Statistics Kingdom, https://www.socscistatistics.com/) were applied to perform the statistical analyses (logistic regression, Fisher’s exact test, Wilcoxon Signed-Rank test), hence the codes are available at the corresponding references. Mutations were further subjected to manual review and categorization, without the use of any codes. The text has been modified (Lines 140-157, 240-245, 276-281, 284-291, 294-297, 386), and data used for the statistical analyses and the corresponding results are now available in Tables S3-S6.

3. Funding statement clarification

Thank you for this important clarification. We have now explicitly included the following statement in our Funding statement and cover letter:

4. Funding information placement

We acknowledge the requirement that funding information must appear exclusively in the Funding Statement section of the online submission form. Accordingly, all funding-related text has been removed from the Acknowledgments. The amended Funding Statement has been included in the cover letter as requested. We respectfully request that the online submission form be updated accordingly.

5. “Data not shown” statement

We acknowledge that the phrase “data not shown” does not meet data sharing standards. All relevant data previously referred to as “data not shown” have now been included in the manuscript (Lines 357-358, 386-389, 508-509).

6. Citation of reviewer-recommended publications

We have carefully evaluated all suggested references. Relevant publications have been incorporated into the revised manuscript where appropriate (statistical analysis: citations No. 36-38; effect of reserpine in Mycoplasma pneumoniae: No. 54). In order to clarify nucleotide positions, the numbering has been given not only according to Escherichia coli, but according to the type strain of Mycoplasma iowae too, as recommended in the review of molecular assays for AMR detection by the MyMIC Consortium (citation No. 39; Lines 135-137, 159-160; Tables S1).

II. Additional Editor Comments

1. Clarification of resistance mechanisms and discussion of unexplained phenotypes

We have clarified in the Materials and Methods and Results sections which resistance mechanisms were examined, including SNP analyses and gene screening. Additionally, we have expanded the Discussion section to explicitly address resistant phenotypes that remain unexplained and have framed these as study limitations. We hope, that the need for further investigation of alternative mechanisms is now clearly acknowledged.

2. Statistical support for SNP–MIC associations

We agree that statistical support strengthens genotype–phenotype interpretation, even if the study first evaluated the nucleotide positions based on literature data about their previous association to resistance. The recommended statistical analyses (including Fisher’s exact test and logistic regression) have now been performed to evaluate SNP–MIC associations. The Results and Materials and Methods sections have been updated accordingly.

3. Moderation of conclusions regarding efflux pump involvement

The conclusions regarding efflux pump involvement have been carefully moderated. We now clearly state the limitations of inhibitor-based assays and avoid overinterpretation of efflux activity based solely on MIC reduction. The Discussion section has been revised to reflect a more cautious and evidence-based interpretation.

4. Alignment of novelty and diagnostic applicability claims

We have carefully revised statements regarding novelty and diagnostic applicability to ensure that they accurately reflect the scope and strength of the data. Claims have been moderated and better aligned with the evidence presented.

III. Reviewer’s Concerns

Major Concerns

1: Incomplete results of resistance mechanisms

We agree that a broader investigation of resistance mechanisms would strengthen the manuscript. However, as - to our knowledge – a first comprehensive investigation in this specific field, our intentions were to assess the presence of known mutations for target modification and develop diagnostic assays for their differentiation. However, Mycoplasma iowae proved to have very unique characteristics among the avian pathogen mycoplasmas, and in the absence of official standards in MIC testing in mycoplasmas, and the detected difference in MIC values in the different media, the description of new associations was beyond the scope of our aims. Thus, we completed our study with the inhibitor-based assay to assess the possibility of involvement of an efflux pump activity in the development of resistance.

Thank You for pointing out the search for tetracycline resistance-associated mutations without reported results. Since the examined strains showed only slightly elevated MIC values - equal or below 4 µg/ml in Oxoid media (the MIC values which were used for the association study) - they were all considered susceptible to oxytetracycline, therefore genetic markers were not searched in the corresponding regions, and accordingly the sentence has been deleted from the Materials and Methods (Lines 128-129, 139-140). For macrolide resistance, results of the analysis of ribosomal proteins (L4 and L22, corrected as appropriate) have now been included (Lines 298-300).

We agree that the findings of four isolates with elevated MICs lacking 23S rRNA position 2059 alterations warrant further investigations, which we now highlight in the discussion (Lines 454-458). As two of these strains were involved in the inhibitor-based assays in case of tilmicosin, we also discuss their phenotype changes caused by orthovanadate in Lines 522-526.

2: Lack of statistical support for genotype–phenotype associations

Following the Reviewer’s recommendations, statistical analyses (Fisher’s exact test and logistic regression) comparing MIC distributions between wild-type and mutant genotypes for gyrA, parC, and 23S rRNA have now been conducted. Statistical methods and results are described in the Materials and Methods and Results sections (Lines 140-157, 276-281, 284-291, 294-297), and data used for the statistical analyses and the corresponding results in details are now available in Tables S3-S6.

3: Limited efflux pump functional validation

We appreciate the valuable suggestions regarding the functional validation of efflux pump activity. Although we are not in the position to perform further experiments, such as the continuation of the studies with reserpine, and transcriptional or protein-level validation was beyond the scope of the current study, we now clearly acknowledge these limitations in the Discussion, and indicate them as important directions for future research. We have also explicitly discussed the limitations of inhibitor-based validation. (Lines 512-518, 528-530)

Minor Comments

Ad 1.

The Abstract has been shortened and streamlined (Lines 35-55). It now clearly states that MAMAs were developed specifically for fluoroquinolone resistance-associated SNPs (Lines 42-44, 53-54).

Ad 2.

The phrase “and were aligned” has been deleted as suggested (Line 262).

Ad 3.

In accordance with the Reviewer’s correction, the discrepancy between the text and Table 2 has been arranged. Primer specificities (including clarification that no allele-specific primer was designed for 248A) are now accurately described and results for the non-specific alleles are presented (Lines 320-329).

Ad 4.

We agree that the interpretation of mixed genotypes is an important practical limitation. The potential for false-negative detection of parC H genotype in mixed populations has now been explicitly discussed in the Discussion section (Lines 488-495).

Ad 5.

Figure 2 has been revised to include all six antimicrobials to provide a comprehensive overview. The figure now shows the log2 transformed MIC values of the antibiotics in Oxoid media, MolliScience, and MolliScience supplemented with sodium orthovanadate. The log2 transformed MIC values of the two-fold dilution series of antimicrobial concentrations are shown to normalize data distribution.

Ad 6.

Table S2 has been updated to include the genomic positions of the efflux pump genes according to the type strain of Mycoplasma iowae, and the text has been completed regarding how regions were analyzed and positions were identified (Lines 263-265). In addition, in order to clarify nucleotide positions in the whole manuscript, the numbering has been given not only according to Escherichia coli, but according to the type strain of Mycoplasma iowae too, as recommended in the review of molecular assays for AMR detection by the MyMIC Consortium (citation No. 39; Lines 135-137, 159-160, Tables S1).

We sincerely thank the Editor and Reviewer for their constructive and thoughtful evaluation. We believe that the revisions have significantly strengthened the scientific accuracy, clarity, and transparency of the manuscript.

Sincerely,

Zsuzsa Kreizinger

---

## [Decision Letter · Decision Letter 1]

16 Mar 2026

Dear Dr. Kreizinger,

Thank you for submitting your manuscript to PLOS ONE. After careful consideration, we feel that it has merit but does not fully meet PLOS ONE’s publication criteria as it currently stands. Therefore, we invite you to submit a revised version of the manuscript that addresses the points raised during the review process.

We look forward to receiving your revised manuscript.

Kind regards,

Chih-Horng Kuo, Ph.D.

Academic Editor

PLOS One

Journal Requirements:

Reviewer's Responses to Questions

**Comments to the Author**

Reviewer #1: (No Response)

2. Is the manuscript technically sound, and do the data support the conclusions?

Reviewer #1: Yes

3. Has the statistical analysis been performed appropriately and rigorously?

Reviewer #1: Yes

4. Have the authors made all data underlying the findings in their manuscript fully available?

Reviewer #1: Yes

5. Is the manuscript presented in an intelligible fashion and written in standard English?

Reviewer #1: Yes

Reviewer #1: The revised version has been significantly improved. There are only a few minor comments to be addressed:

1. Figure 2. Showing the MIC value bars in the negative direction for those less than 1 is somewhat strange. Maybe try to use a broken/split axis bar chart instead of using the log2 transformation.

2. One strain with A2059G mutation in 23S rRNA is susceptible to erythromycin. If this is not due to variant calling error, it would be interesting to provide an explanation.

3. The MAMA primer gyrA248-H specificity has been clarified to be only for C248T. The C248A information in M. iowae reference should be excluded also (lines 322, 468, 480, and table 3).

.

Reviewer #1: No

---

## [Author Response · Author response to Decision Letter 2]

30 Mar 2026

Zsuzsa Kreizinger

HUN-REN Veterinary Medical Research Institute

Hungária krt. 21.

H-1143 Budapest, Hungary

Phone: +36 1 467 4060 Fax: +36 1 467 4076

E-mail: kreizinger.zsuzsa@vmri.hun-ren.hu

Chih-Horng Kuo

Academic Editor

PLOS One

30th March, 2026

Dear Chih-Horng Kuo,

We thank You and the Reviewer for Your e-mail on 17th of March, 2026 and for Your careful evaluation of our manuscript” Identification and detection of genetic markers associated with antimicrobial susceptibility and evaluation of efflux pump mechanisms in Mycoplasma iowae” and for the constructive comments provided.

I. Response to Journal Requirements

Citations

We have carefully evaluated all suggested references during the previous revision, no new recommendations were added. Relevant publications have been incorporated into the revised manuscript where appropriate (statistical analysis: citations No. 36-38; effect of reserpine in Mycoplasma pneumoniae: No. 54). In order to clarify nucleotide positions, the numbering has been given not only according to Escherichia coli, but according to the type strain of Mycoplasma iowae too, as recommended in the review of molecular assays for AMR detection by the MyMIC Consortium (citation No. 39).

All statistical analyses were repeated or performed according to the first revision of the manuscript using the cited tools, therefore the previously cited software (R 4.3.3) has been eliminated from the citation list.

A full audit with publisher-level verification was conducted using ChatGPT to assess the reference list for retractions. This included checking direct sources (PubMed and official publisher pages) for retraction status, reviewing PubMed retraction flags, examining publisher platforms (Elsevier, Taylor & Francis, Springer, Frontiers, PLoS), and performing Retraction Watch–type searches using title and author combinations. No retracted publications or linked retraction notices were identified in our reference list.

II. Reviewer’s Comments

Ad 1.

Figure 2 has been adjusted and now split axis is used instead of log2MIC bars to better visualize MIC values, the description of the figure has been corrected accordingly (Fig2, Lines 392, 396-397).

Ad 2.

In accordance with the Reviewer’s comment, we completed our discussion about unexplained resistant phenotypes as limitations with the unexplained susceptible phenotype in case of macrolides and lincomycin (Lines 455-457).

Ad 3.

According to the clarified MAMA specificity, the text has been corrected when the MAMA-targeted mutation was mentioned (Lines 322, Table 3). The discussion details examinations on the sequences and MIC values, not on the MAMA results, therefore in these cases, the “C248T/A” form was considered appropriate (Lines 469, 481). Nevertheless, the specificity of the gyrA-MAMA has been addressed in the discussion also (Lines 484-485).

We thank the Editor and Reviewer for their evaluation. We consider that the revisions have improved the scientific accuracy, clarity, and transparency of the manuscript.

Sincerely,

Zsuzsa Kreizinger

---

## [Editor Report · Decision Letter 2]

31 Mar 2026

Identification and detection of genetic markers associated with antimicrobial susceptibility and evaluation of efflux pump mechanisms in Mycoplasma iowae

PONE-D-25-63087R2

Dear Dr. Kreizinger,

We’re pleased to inform you that your manuscript has been judged scientifically suitable for publication and will be formally accepted for publication once it meets all outstanding technical requirements.

Kind regards,

Chih-Horng Kuo, Ph.D.

Academic Editor

PLOS One
---

## [Editor Report · Acceptance letter]

PONE-D-25-63087R2

PLOS One

Dear Dr. Kreizinger,

I'm pleased to inform you that your manuscript has been deemed suitable for publication in PLOS One. Congratulations! Your manuscript is now being handed over to our production team.

Kind regards,

on behalf of

Dr. Chih-Horng Kuo

Academic Editor

PLOS One